# Anatomic Characterization of the Ocular Surface Microbiome in Children

**DOI:** 10.3390/microorganisms7080259

**Published:** 2019-08-14

**Authors:** Kara M. Cavuoto, Anat Galor, Santanu Banerjee

**Affiliations:** 1Bascom Palmer Eye Institute, University of Miami Miller School of Medicine, 900 NW 17th Street, Miami, FL 33136, USA; 2Miami Veterans Administration Medical Center, 1201 NW 16th Street, Miami, FL 33125, USA; 3Department of Surgery, University of Miami Miller School of Medicine, 1295 NW 14th Street, Miami, FL 33136, USA

**Keywords:** microbiome, ocular surface, children, composition, eyelid, skin

## Abstract

The microbiome is important in the evolution of the immune system in children; however, information is lacking regarding the composition of the pediatric ocular microbiome and its surrounding structures. A prospective, cross-sectional study of the ocular microbiome was conducted in children <18 years old. Samples from the inferior conjunctival fornix of both eyes, eyelid margin, and periocular skin underwent DNA amplification and 16S sequencing using Illumina MiSeq 250. The microbiome was analyzed using Qiime. Statistical analysis was performed using a two-sided Student’s *t*-test, diversity indices, and principal coordinate analysis. A total of 15 children were enrolled. The ocular surface microbiome was predominantly composed of Proteobacteria, whereas Bacteroidetes dominated the eyelid margin, and Firmicutes dominated the periocular skin. Despite these variations, no statistically significant compositional differences were found with Bray-Curtis analysis. The conjunctiva had the lowest Shannon diversity index with a value of 2.3, which was significantly lower than those of the eyelid margin (3.4, *p* = 0.01) and the periocular skin (3.5, *p* = 0.001). However, the evenness of the species using Faith’s phylogenetic diversity index was similar at all sites. Overall, the ocular surface microbiome is dominated by Proteobacteria in children. The niche is similar to the surrounding structures in terms of composition, but has a lower number and relative abundance of species.

## 1. Introduction

The microbiome is the collection of organisms that live on and in us [1]. The diversity and abundance of bacteria are determined by the anatomic niche in which they live [2]. These bacteria are important, as they contribute to vital functions, including nutritional, developmental, defensive, and physiologic processes [3]. Therefore, the composition and evolution of the microbiome have important implications for health and disease [4].

The timing of microbial colonization is not clearly defined. Initially, it was thought that the first bacteria are introduced at birth; however, the current theory is that colonization begins in utero [5]. These bacteria are hypothesized to originate from the mother’s genitourinary tract and/or via hematogenous transmission from the placenta [6,7]. At birth, there is a drastic change in the abundance and types of bacteria. Initial studies looking at conjunctival cultures performed within 15 min of vaginal birth found that the Lactobacillus species, the primary colonizer of the female reproductive tract, accounted for more than half of all bacteria isolated [8,9]. More recent studies found that the change in bacterial flora is determined by the method of birth, as there were differences based on the mode of delivery. Not only was the proportion of positive conjunctival cultures obtained within 1 h after birth higher in infants born vaginally compared to those born via Cesarean section, but cultures from vaginally-delivered infants also showed more bacterial strains and a greater number of different species per patient than those from infants born via Cesarean section [10]. Interestingly, when conjunctival cultures were obtained 2 days after birth and compared with those obtained on the day of delivery, they were more frequently positive regardless of the method of delivery [11]. It is likely that this evolution of the microbiome continues in the weeks and months after birth, as influences such as geographic location, diet, and environmental exposures further modulate the microbiome’s composition.

While the compositions of the gut and lung microbiomes have been extensively studied, little is known about the ocular surface microbiome in children. Prior studies have demonstrated that the ocular surface microbiome is paucibacterial compared to bacteria-rich niches, such as the skin or gastrointestinal tract [12]. Additionally, while studies have shown that the ocular surface microbiome is composed primarily of Firmicutes, Proteobacteria, and Actinobacteria, the majority of studies only analyzed adults or did not examine children independently [12,13]. We have previously examined this question and found differences in the microbiome between children and adults [14] and between older and younger children [15], but we do not know how the composition of the conjunctival microbiome in children differs from that of the surrounding ocular structures.

To fill in this knowledge gap, we conducted a prospective study to characterize the ocular microbiome in children <18 years old by analyzing the conjunctiva, eyelid margin, and periocular skin of the cheek using 16S sequencing.

## 2. Methods

### 2.1. Study Design, Setting, and Population

A prospective, cross-sectional, observational study of children <18 years old at a university-based institution was conducted with approval by the University of Miami Institutional Review Board (ID 20140717, 12 December 2014) and in accordance with the principles of the Declaration of Helsinki. All children (<18 years of age) scheduled for a visit to the pediatric ophthalmology clinic were invited to participate. Patients were excluded if they had a current ocular or intraocular infection, had used antibiotics (topical or oral) within the prior 30 days, had undergone ocular surgery within the prior 90 days, or the patient or guardian refused participation. Written informed consent was obtained from parents/guardians, and written informed assent was obtained from children aged 7–17 years. The methodology was performed in accordance with a previously published study by our group [14] as detailed below.

### 2.2. Specimen Collection

Specimens were collected with a forensic-quality applicator without anesthetic from the following sites: right inferior conjunctival fornix, left inferior conjunctival fornix, right eyelid margin, and right periocular skin. The collected specimens were then immediately transported by study personnel directly to the microbiology laboratory and stored in a −80 °C freezer.

### 2.3. 16S Sequencing

Swab heads were aseptically transferred into PowerSoil sample collection tubes and lysed using a MagnaLyser tissue disruptor (Roche, Indianapolis, IN, USA), and total DNA was isolated using a PowerSoil/Fecal DNA isolation kit (Mo-Bio, Germantown, MD, USA) as per manufacturer’s specifications. All samples were quantified using the Qubit^®^ Quant-iT dsDNA Broad-Range Kit (Life Technologies, Grand Island, NY, USA) to ensure that they met minimum concentration and mass of DNA, and were submitted to University of Minnesota Genomic Center for 16S rRNA sequencing. The DNA was amplified using fusion primers designed against the surrounding conserved regions that are tailed with sequences to incorporate flow cell adapters and indexing barcodes (Illumina, San Diego, CA, USA). Each sample was polymerase-chain-reaction (PCR) amplified with two differently barcoded V4–V5 fusion primers and was advanced for pooling and sequencing. For each sample, amplified products were concentrated using a solid-phase reversible immobilization method for the purification of PCR products and quantified by electrophoresis using a 2100 Bioanalyzer (Agilent, Santa Clara, CA, USA). The pooled 16S V4–V5-enriched, amplified, barcoded samples were loaded into the MiSeq cartridge (Illumina Inc, San Diego, CA, USA), and then onto the instrument along with the flow cell. After cluster formation on the MiSeq Instrument (Illumina, San Diego, CA, USA), the amplicons were sequenced for 250 cycles with custom primers designed for paired-end sequencing.

Unused swabs and reagent controls were supplied in triplicate as background. DNA extraction reagents and blank sampling tools acted as negative controls, and these were carried along with the ocular samples through the same pipeline of DNA extraction to QIIME analysis. The Operational Taxonomic Unit (OTU) file revealed several unique counts for each blank tool and a few with minor overlap with ocular samples. To discount the possibility of implement-induced artifacts, OTUs with comparable representation in the blank samples were eliminated from further analysis to derive a highly conservative OTU list. Samples producing amplicons at later cycles compared with the majority of samples were concentrated using Agencourt AMPureXP beads (Beckman Coulter, Indianapolis, IN, USA). All samples were sequenced together after barcode normalization subsequent to a preliminary sequencing run. Microbiome analysis sequences were quality filtered and demultiplexed using exact matches to the supplied DNA barcodes and primers. The resulting sequences were then searched against the SILVA release 123 database of 16S sequences, clustered at 99% (closed-reference Operational Taxonomic Unit (OTU) picking) to obtain phylogenetic identities.

### 2.4. Statistical and Bioinformatics Analysis

Microbiome analysis was performed using QIIME 2. OTU tables were rarefied to the sample containing the lowest number of sequences in each analysis. First, the microbiome composition was analyzed overall and at the four distinct anatomic sites (conjunctiva of the right and left eye, eyelid margin of the right eye, and periocular skin of the right eye). The diversity of the microbiome was then analyzed using Shannon index and Faith’s phylogenetic diversity. Shannon index (H) was used to characterize the overall species diversity. This index takes into account both the number of species present and the relative abundance of each species, demonstrating the number of different OTUs encountered and the instances these unique OTUs were sampled. An increase in the number of different OTUs and instances indicates a higher diversity (a larger H). The general range for the Shannon index (H) found in ecological studies is typically between 1.5 and 3.5, and generally never greater than 4 [16]. Faith’s phylogenetic diversity was used to estimate the biodiversity of a collection of species, taking into account the phylogenetic overlap among taxa. Beta diversity was qualitatively examined using Bray-Curtis principal coordinate analysis. Permutational multivariate analysis of variance and Kruskal-Wallis were utilized for finding significant whole-microbiome differences between discrete categorical, discontinuous, or continuous variables. The fraction of permutations with greater distinction between categories than that observed with the non-permuted data was reported as the p-value. All other data forms were analyzed for significance with the Mann–Whitney U-test (GraphPad Prism).

## 3. Results

16S sequencing was performed on 30 conjunctival samples (15 right eye, 15 left eye), 15 eyelid margin samples (right eye), and 15 periocular skin samples (right cheek) from 15 children <18 years old (mean 3.7 years, standard deviation ± 31 months). The number of distinct operational taxonomic units (OTUs) was 359. The conjunctiva demonstrated the lowest number of OTUs with an average of 122 distinct OTUs (Figure 1). The number of OTUs in the conjunctiva was similar between the right and left eyes. The periocular skin had the highest number of OTUs (172) among the sites sampled.

At the phylum level, the composition of the conjunctival microbiome was dominated by Proteobacteria (57%), followed by Firmicutes (17%), Bacteroidetes (13%), and Actinobacteria (11%) (Figure 2a). This differed from the composition of the eyelid margin, in which Bacteroidetes (41%) was the dominant phylum, followed by Firmicutes (39%) and Proteobacteria (13%) (Figure 2b). The periocular skin composition also differed from that of the conjunctiva in that it was primarily composed of Firmicutes (45%) and then to lesser amounts of Proteobacteria (35%), Bacteroidetes, (13%) and Actinobacteria (5%) (Figure 2c). The taxa bar plot demonstrating the relative abundance at the phylum level is shown in Figure 3. Despite these qualitative differences, no significant differences in composition were found between the right and left eye [Figure 4a] or between the conjunctiva, eyelid margin, and periocular skin (Figure 4b) when the diversity of microbial composition (beta diversity) was examined with Bray-Curtis principal coordinate analysis (pCoA). This indicates that the overall number of different species was similar in all the sites tested.

Diversity analysis encompasses both the number of species in the sample (richness) and the relative abundance of species (evenness) [16]. As the Shannon index includes both of these features, the microbiome was first analyzed using this index. The conjunctiva had the lowest Shannon index with an average H of 2.3, whereas the eyelid margin and skin both showed greater diversity with a higher H of 3.4 and 3.5, respectively (Figure 5a) The difference in the Shannon index was significant between the conjunctiva and eyelid margin (*p* = 0.01), and between the conjunctiva and periocular skin (*p* = 0.001); however, it was not statistically different when comparing the eyelid margin and the periocular ocular skin (*p* = 0.26).

Faith’s phylogenetic diversity measurement was performed to analyze the biodiversity of a collection of species. There was no significant difference between the conjunctiva of the right and left eye (*p* = 0.52, Kruskal-Wallis). Additionally, there were no significant differences between the conjunctiva and eyelid margin (*p* = 0.90), margin and periocular skin (*p* = 0.51), or conjunctiva and periocular skin (*p* = 0.30) (Figure 5b). This indicates that the relatedness of the species was not drastically different when comparing both eyes or when comparing any of the anatomic sites. Although initially this may seem contradictory to the results from the Shannon diversity index, it is plausible, as phylogenetic diversity reflects the sum of all branch lengths of the OTUs in the group. Two groups may have different Shannon diversity indices but similar Faith indices if the number of branches they contribute is similar on the phylogenetic tree [16].

## 4. Discussion

Overall, we found that the pediatric ocular surface microbiome was paucibacterial when compared with the eyelid margin and skin, with a composition that consisted primarily of Proteobacteria, Firmicutes, and Bacteroidetes. All three sites were paucibacterial compared with the gastrointestinal tract, where >2000 OTUs are routinely recovered [17]. The conjunctiva, besides having fewer organisms, had also a lower diversity in terms of richness and evenness when compared to the eyelid margin and surrounding skin. However, neither the relatedness of the species nor the number of distinct species differed significantly when comparing the conjunctiva, eyelid margin, and periocular skin. This would imply that although there were some differences identified, overall, the microbiome of the ocular surface is similar to those of neighboring structures in children.

There is little information in the literature on the composition of the pediatric ocular surface microbiome. In a previous study, we found 551 OTUs in 50 patients <18 years old in specimens collected with the same protocol and from same patient population [15]. The most abundant organisms were Proteobacteria, Firmicutes, and Bacteroidetes, similar to the present study. However, it is unclear why our most recent study found fewer OTUs. Explanations for this finding include a more stringent filtering strategy (i.e., confidence interval of 99% instead of 97%) and a slightly older age group (mean age of 44 months vs. 36 months). A study from Gambia included 105 samples from healthy conjunctiva of both children (*n* = 21) and adults (*n* = 84), and the ocular surface microbiome was composed of Actinobacteria (46%), Proteobacteria (24%), and Firmicutes (22%) [18]. However, the authors in that study did not report the composition of the pediatric microbiome separately from the adult microbiome, and did not report the number of OTUs. Additionally, they used the V1–V3 region of the bacterial 16S rRNA to characterize the microbiome, whereas we used the V4–V5 regions, which may also account for some of the differences. These variations, in addition to the geographic location, may have contributed to the differences seen.

More data exists on the ocular surface microbiome of adults than children. Studies with adults found that the number of OTUs in ocular surface samples ranges from 159 to 221 in healthy individuals [19,20,21], which is slightly higher than in our study. In terms of composition, our study agrees that Proteobacteria was the most abundant phylum on the ocular surface. However, while we found Firmicutes and Bacteroidetes were the next most abundant in children, studies with adults found Actinobacteria and Firmicutes were the next two most abundant phyla [14,19,20]. These findings would suggest that there is an age-related evolution of the ocular surface microbiome, but more studies are needed to directly address this question.

In addition to age, anatomic site seems to play a role in the composition of the microbiome. In the current study, children demonstrated an overlap between the conjunctival, eyelid margin, and skin microbiomes, whereas the previous literature has shown that these niches are distinct in adults [12,14,21]. Our results showed that the eyelid margin appears to be an intermediary zone, with a relative abundance of Firmicutes and OTU counts between those of the conjunctiva and skin. A possible explanation for lower OTUs in the conjunctiva is that the ocular surface defense systems, including physical barriers such as tears, mechanical barriers such as blinking, and immunologic barriers such as IgA and lactoferrin, may limit the number of organisms on the ocular surface [22].

As with all studies, our findings should be considered in light of several limitations. First, 16S sequencing is prone to noise, sampling errors and contamination. However, we included blank specimens that went through DNA extraction and QIIME analysis and acted as negative controls, and we subsequently eliminated any OTUs found in blank samples from further analysis. Second, our sample size was small, given that this was a pilot study to demonstrate the ability to characterize the pediatric ocular microbiome within an individual. This small number of patients would not allow controlling for or analyzing the multitude of various factors that may affect the microbiome, including gender and antimicrobial exposure. Despite the small number of patients, we successfully obtained and analyzed specimens from all patients, and demonstrated the reliability of our methods, as the findings were similar compared with our prior publication and those of other authors. Third, the patients presenting for care at our institution are representative of the surrounding community; however, our patient population may not be representative of the general population, particularly at different geographic sites. Future studies recruiting different populations across the United States and on different continents could be pursued in order to more accurately assess the influence of the immediate environment on the ocular surface microbiome. Finally, the cross-sectional nature of the data allows for a snapshot in time of the microbiome, but cannot demonstrate how the microbiome evolves over time, what events or exposures precipitate change, the duration of the exposure required to exert a change, or the durability of change. However, it has been demonstrated that the microbiome is relatively stable within an individual over time [13].

Despite these limitations, our data adds to the growing understanding of the ocular microbiome, which may differ in children and adults. Our study confirmed that the composition is similar between both eyes, with contributions primarily from Proteobacteria, Firmicutes, and Bacteroidetes. We hypothesize that the conjunctiva is tightly regulated by the ocular defense systems, as suggested by the lowest OTUs and Shannon index. Future studies will be needed to understand the evolution in an individual and whether manipulation of the ocular surface microbiome will have effects on health and disease.

## Figures and Tables

**Figure 1 microorganisms-07-00259-f001:**
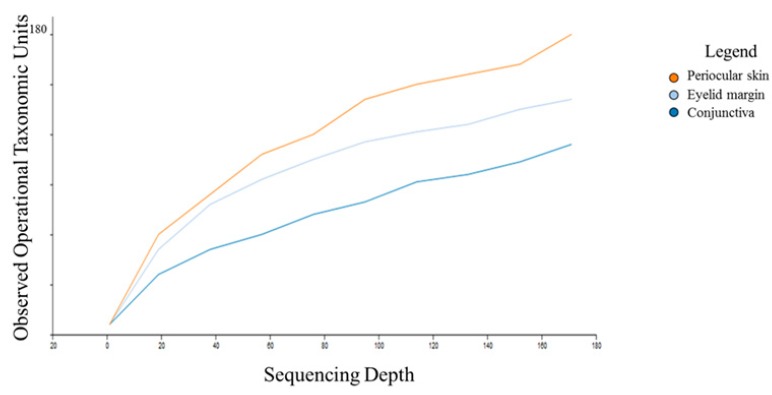
Number of operational taxonomic units by anatomic site.

**Figure 2 microorganisms-07-00259-f002:**
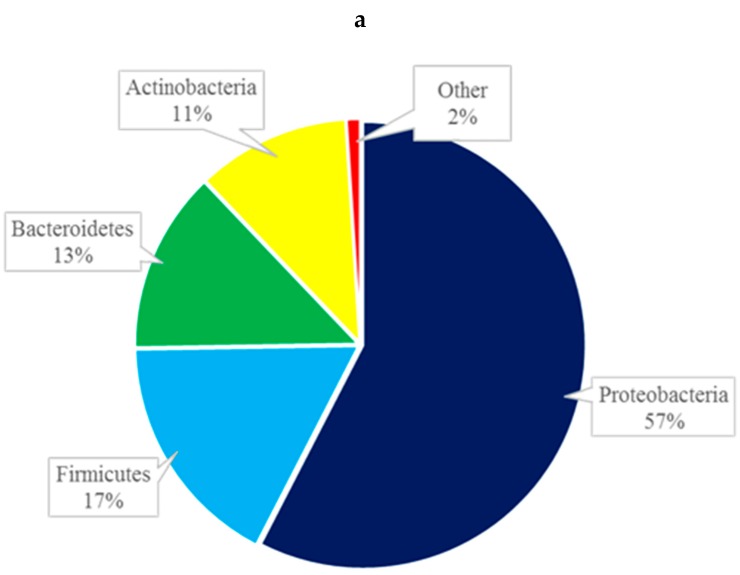
Composition of the microbiome by phylum of the conjunctiva (**a**), eyelid margin (**b**), and periocular skin (**c**).

**Figure 3 microorganisms-07-00259-f003:**
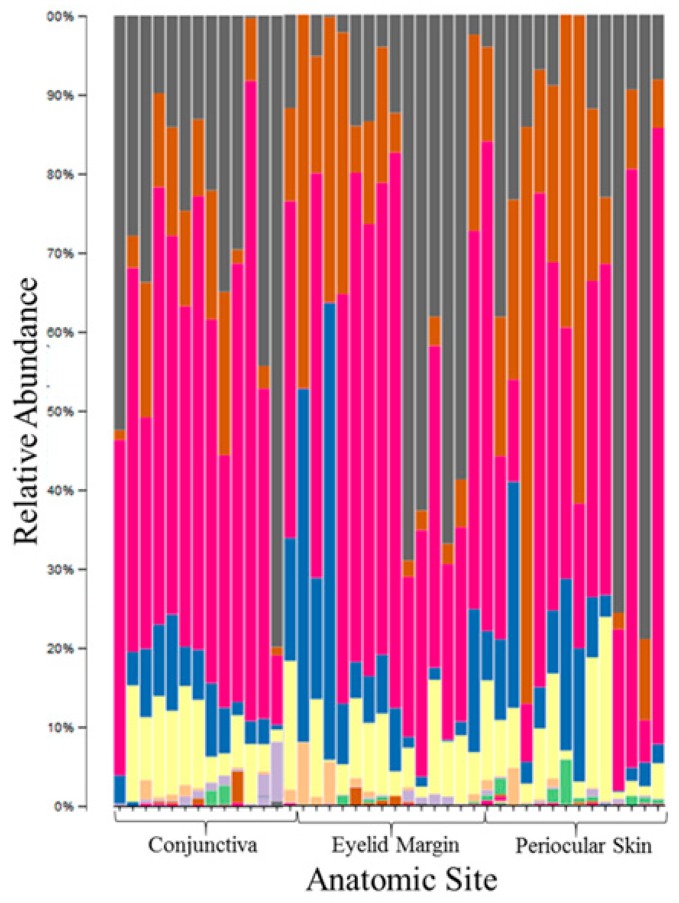
Taxa bar plot of the relative abundance at each anatomic site at the phylum level.

**Figure 4 microorganisms-07-00259-f004:**
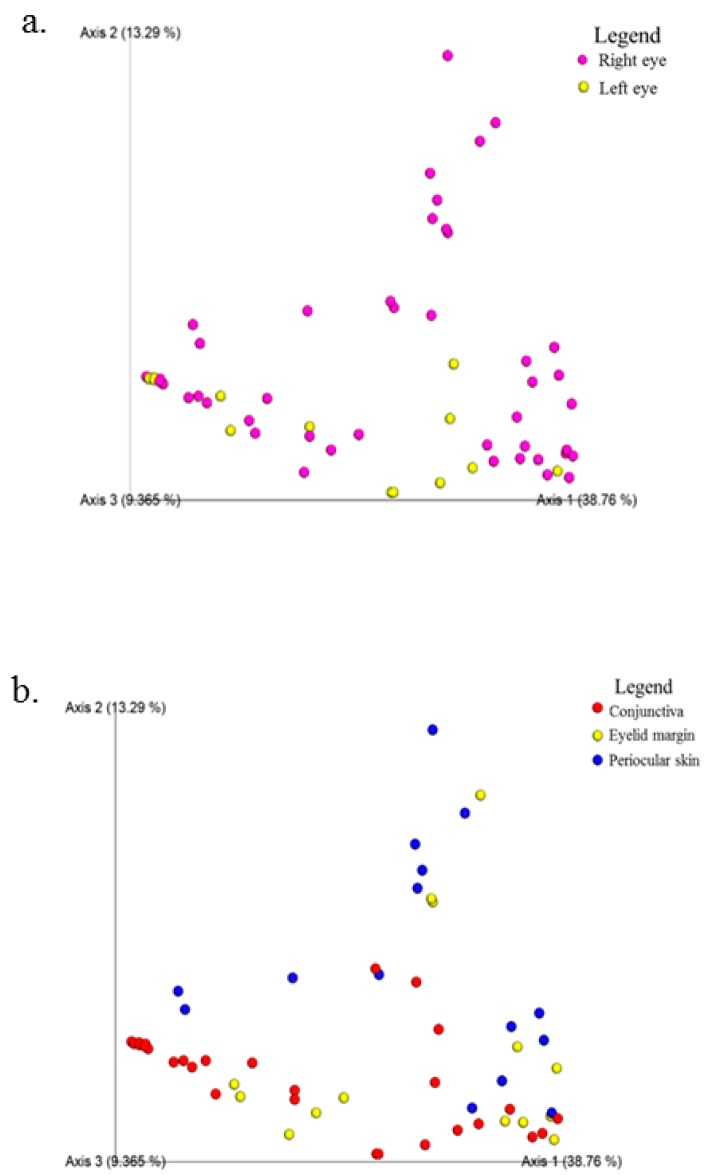
Beta diversity demonstrated with Bray-Curtis principal coordinate analysis. (**a**) Comparison of the right and left conjunctiva. (**b**) Comparison of the conjunctiva, eyelid margin, and periocular skin.

**Figure 5 microorganisms-07-00259-f005:**
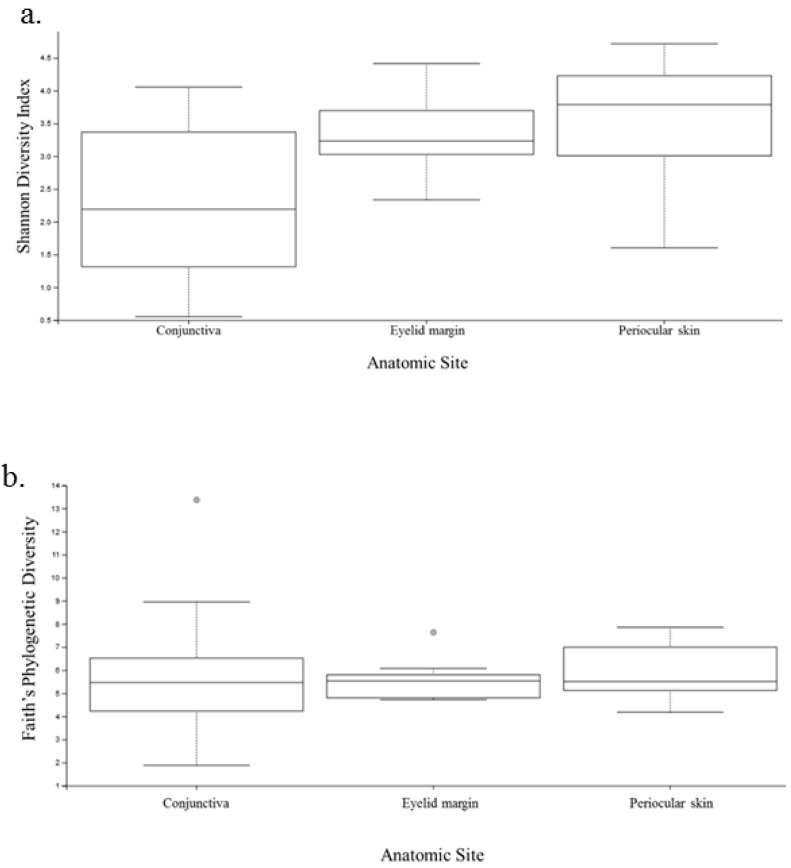
Plots of the diversity indices comparing the conjunctiva, eyelid margin, and periocular skin. (**a**) Shannon index comparing the three anatomic sites. (**b**) Faith’s phylogenetic diversity comparing the three anatomic sites.

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
