# Peer review of "Anatomic Characterization of the Ocular Surface Microbiome in Children"

_microorganisms, 2019, doi:10.3390/microorganisms7080259_

Round 1

Reviewer 1 Report

This study is described as a study of children younger than 18 years of age. I don't know that i would describe a 15-17 year old as a child. The age range seems to me to be a little wide. I would like to see narrower age ranges with a larger N value. Nevertheless, i agree with the authors on the paucity of data on the ocular microbiome of children and this work is fine as preliminary research.

Author Response

August 11, 2019

Dear Reviewer 1:

Thank you for your time and effort in the review of our manuscript entitled “Anatomic Characterization of the Ocular Surface Microbiome in Children” for publication in Microorganisms.

Point 1: This study is described as a study of children younger than 18 years of age. I don't know that i would describe a 15-17 year old as a child. The age range seems to me to be a little wide. I would like to see narrower age ranges with a larger N value. Nevertheless, i agree with the authors on the paucity of data on the ocular microbiome of children and this work is fine as preliminary research.

Response 1: The authors appreciate the Reviewer’s points. We agree that the age range is wide, however we sought to keep our definition of “child” in line with the National Institutes of Health’s definition as < 18 years old.  We agree that a larger N would be beneficial in further defining the microbiome in this group of patients. Future studies will be performed to address this point.

Thank you for your consideration. Please do not hesitate to contact us for any questions or further clarifications.

Sincerely,

Kara Cavuoto, MD

Reviewer 2 Report

It is an interesting paper to show that  the ocular surface  and adjacent structures share similarities in the microbiomes in children. Even though there are some limitations of the study, such as small sample size andlack of regional diversity, this paper still shows some novelty. 

Author Response

Dear Reviewer 2:

Thank you for your time and effort in the review of our manuscript entitled “Anatomic Characterization of the Ocular Surface Microbiome in Children” for publication in Microorganisms.

Point 1: It is an interesting paper to show that  the ocular surface  and adjacent structures share similarities in the microbiomes in children. Even though there are some limitations of the study, such as small sample size andlack of regional diversity, this paper still shows some novelty.

Response 1: The authors appreciate the Reviewer’s points and agree that the limitations include small sample size and lack of regional diversity. Despite these limitations, we appreciate the opportunity to publish our manuscript.

Thank you for your consideration. Please do not hesitate to contact us for any questions or further clarifications.

Sincerely,

Kara Cavuoto, MD